# Prothrombin Time-International Normalized Ratio Predicts the Outcome of Atrial Fibrillation Patients Taking Rivaroxaban

**DOI:** 10.3390/biomedicines10123210

**Published:** 2022-12-10

**Authors:** Tze-Fan Chao, Yi-Hsin Chan, Pei-Chien Tsai, Hsin-Fu Lee, Shang-Hung Chang, Chi-Tai Kuo, Gregory Y. H. Lip, Shih-Ann Chen, Yung-Hsin Yeh

**Affiliations:** 1Division of Cardiology, Department of Medicine, Taipei Veterans General Hospital, Taipei 11217, Taiwan; 2Cardiovascular Research Center, Institute of Clinical Medicine, National Yang Ming Chiao Tung University, Taipei 11221, Taiwan; 3The Cardiovascular Department, Chang Gung Memorial Hospital, Taoyuan 33305, Taiwan; 4College of Medicine, Chang Gung University, Taoyuan 33302, Taiwan; 5Microscopy Core Laboratory, Chang Gung Memorial Hospital, Taoyuan 33305, Taiwan; 6Department and Graduate Institute of Biomedical Sciences, Chang Gung University, Taoyuan 33302, Taiwan; 7Division of Pediatric Infectious Diseases, Department of Pediatrics, Chang Gung Memorial Hospital, Taoyuan 33305, Taiwan; 8Graduate Institute of Clinical Medical Sciences, College of Medicine, Chang Gung University, Taoyuan 33302, Taiwan; 9Center for Big Data Analytics and Statistics, Chang Gung Memorial Hospital, Taoyuan 33305, Taiwan; 10Liverpool Centre for Cardiovascular Science at University of Liverpool, Liverpool John Moores University and Liverpool Heart & Chest Hospital, Liverpool L14 3PE, UK; 11Department of Clinical Medicine, Aalborg University, 9100 Aalborg, Denmark

**Keywords:** atrial fibrillation, DOACs, prothrombin time, activated partial thromboplastin time, ischemic stroke, major bleeding

## Abstract

Background: Although direct oral anticoagulants (DOACs) for patients with atrial fibrillation (AF) are considered to be safe, over or under anticoagulation and increased bleeding or thromboembolic risk are still considered individually. We aimed to investigate whether there is an association between prothrombin time and international normalized ratio (PT-INR) or activated partial thromboplastin time (aPTT) ratio, and the risks of ischemic stroke/systemic embolism (IS/SE) and major bleeding among AF patients taking rivaroxaban or dabigatran. Methods: This multi-center cohort study in Taiwan included 3192 AF patients taking rivaroxaban and 958 patients taking dabigatran for stroke prevention where data about PT-INR and aPTT were available. Results: For patients treated with rivaroxaban, a higher INR level was not associated with a higher risk of major bleeding compared to an INR level < 1.1. The risk of IS/SE was lower for patients having an INR ≥ 1.5 compared to those with an INR < 1.1 (aHR:0.57; [95%CI: 0.37–0.87]; *p* = 0.01). On-label dosing of rivaroxaban and use of digoxin were independent factors associated with an INR ≥ 1.5 after taking rivaroxaban. For patients taking dabigatran, a higher aPTT ratio was not associated with a higher risk of major bleeding. The risk of IS/SE was lower for patients having an aPTT ratio of 1.1–1.2 and 1.3–1.4 than those with an aPTT ratio < 1.1. Conclusions: In AF patients, rivaroxaban with an INR ≥ 1.5 was associated with a lower risk of IS/SE. PT-INR or aPTT ratios were not associated with bleeding events for rivaroxaban or dabigatran. INR may help predict the outcome of AF patients who take rivaroxaban.

## 1. Introduction

Atrial fibrillation (AF) is associated with an increased risk of systemic thromboembolism, heart failure, and mortality [1]. Current guidelines recommend direct oral anticoagulants (DOACs) (e.g., dabigatran, rivaroxaban, apixaban, and edoxaban) rather than vitamin K antagonists (VKAs) (e.g., warfarin) as first-line therapies for stroke prevention in AF patients [2]. Frequent measurements of prothrombin time-international normalized ratio (PT-INR) and subsequent dosage adjustments when indicated were essential for VKA users [3]. In contrast, the routine coagulation testing is generally not required for DOACs since their pharmacodynamic and pharmacokinetic properties are more predictable. Indeed, four large-scale randomized controlled studies (RCTs) including the RE-LY, ROCKET AF, ARISTOTLE, and ENGAGE AF-TIMI 48 have demonstrated that DOACs at a fixed dose, according to predefined dosage criteria without blood tests for coagulation profiles or drug concentrations, were at least as effective in risk of thromboembolism and safer in terms of risk of major bleeding when compared to warfarin [2]. However, a high interindividual variability in plasma drug levels of DOACs was found, and the Phase III trials found a positive association between DOAC concentration and major bleeding during follow up [4,5].

The anticoagulant effect of the FXa inhibitors (e.g., rivaroxaban, apixaban and edoxaban) can be quantitatively monitored by using the Anti-FXa chromogenic assays with validated calibrators. Conversely, the diluted thrombin time (dTT) displays a direct linear relationship with dabigatran concentration and is suitable for the quantitative assessment of dabigatran concentrations. In contrast to the routine coagulation tests, such as PT-INR or activated partial thromboplastin time (aPTT), the monitoring of anti-FXa activity is not widely available in clinical practice. Previous studies indicated that FXa inhibitors were associated with a concentration-dependent prolongation of the PT [6,7,8]. PT-INR is rapidly available and may help the clinicians evaluate the possibility of recent exposures of FXa-inhibitors in several emergent conditions, such as urgent procedures, major bleeding, or an acute stroke [2,9]. For dabigatran, the aPTT may provide a quantitative assessment of dabigatran serum level and anticoagulant activity. The relationship between dabigatran serum concentration and the aPTT value followed a curvilinear curve with a quadratic curve [2,7,8]. However, previous studies investigating whether the measurements of PT-INR or aPTT could be useful for monitoring or correlating with the concentration of rivaroxaban and dabigatran have showed conflicting results [7,8,10,11,12,13].

Because the INR or aPTT ratio value of 1.5 is commonly used as a cut-off threshold to determine the effective anticoagulant activity in patients treated with warfarin (INR value of 1.5 for Asians specifically) or heparin, [14,15] we aimed to investigate whether there was an association between the PT-INR or aPTT ratio (e.g., also using the cut-off value of 1.50), and the risks of ischemic stroke/systemic embolism (IS/SE) and major bleeding among AF patients taking rivaroxaban or dabigatran, respectively. We hypothesized that a higher INR or aPTT ratio was associated with a lower risk of thromboembolism and/or was associated with a higher risk of major bleeding among AF population treated with rivaroxaban or dabigatran, respectively.

## 2. Materials and Methods

This present study was approved by the Institutional Review Board of the Chang Gung Medical Foundation. In the retrospective cohort study, we used data from the Chang Gung Memorial Hospital Medical System, which is the largest healthcare provider in Taiwan with approximately 1/10 of the Taiwanese medical service annually [16]. The advantage of the CGMH medical database is that each patient’s detailed chart record, diagnosis, imaging, and laboratory data are available. The identification number and personal information of each patient are encrypted and de-identified by using a consistent encrypting procedure; therefore, informed consent was waived for this study. The interpretation and conclusions contained herein do not represent the position of Chang Gung Memorial Hospital.

### 2.1. Study Design

The flowchart of study design and patient enrollment is shown in Figure 1. From 1 June 2012 to 30 September 2018, a total of 5541 and 2274 patients diagnosed with non-valvular AF taking rivaroxaban or dabigatran were identified. AF was diagnosed using the International Classification of Diseases (ICD), Ninth Revision, Clinical Modification (ICD-9-CM) (427.31) or ICD-10-CM (I48) codes registered by the physicians responsible for the treatments of patients. To establish a cohort of non-valvular AF patients taking OACs for the primary purpose of stroke prevention, patients were excluded if any diagnosis indicating valvular AF (mitral stenosis or history of valvular surgery), venous thromboembolism (deep vein thrombosis or pulmonary embolism) or joint replacement therapy within 6 months before DOAC prescriptions was present. Only those patients with at least one time of PT-INR or aPTT measurement after taking DOACs were enrolled in the present study. For those patients with more than one time of INR or aPTT measurement after treating with DOAC, we adopted the first INR or aPTT measurement after taking DOAC more than one week. The follow-up period was defined as the duration from the drug index date until the occurrence of study outcomes or until the end date of the study period (30 September 2018), whichever came first. The study finally identified a total of 3192 patients taking rivaroxaban with PT-INR available and 958 patient taking dabigatran with aPTT available, respectively.

### 2.2. Study Outcomes

The clinical outcomes of the present study were the occurrences of hospitalized IS/SE and major bleeding. Hospitalized major bleeding events were defined as hospitalization due to symptomatic bleeding in a critical organ including brain, gastrointestinal tract, other critical sites, or fatal bleeding. All study outcomes were defined on the basis of the first three discharge diagnoses to avoid misclassifications.

### 2.3. Covariates

Baseline covariates referred to any claim record with the above diagnoses, or medication codes prior to the index date. Bleeding history was confined to events within 6 months preceding the index date. A history of any prescription medicine was confined to medications taken at least once within 3 months preceding the index date. Important laboratory data, including serum hemoglobin, platelet count, estimated glomerular filtration rate (eGFR), and alanine aminotransferase (ALT), were based on the measurements performed within 1 year of the index date. The CHA_2_DS_2_-VASc score (congestive heart failure, hypertension, age 75 years or older for 2 points, diabetes mellitus, previous stroke or transient ischemic attack for 2 points, vascular disease, age 65 to 74 years, and female gender) was computed to represent the predicted risk of IS/SE and the HAS-BLED score (hypertension, abnormal renal/liver function, stroke, bleeding history, labile INR, age 65 years or older, and antiplatelet drug/alcohol use) was adopted to predict the risk of major bleeding in patients with non-valvular AF treated with OACs [17,18].

### 2.4. Statistical Analysis

Data were presented as mean values with standard deviations or median values with interquartile ranges for continuous variables, and as proportions for categorical variables. χ^2^ test was used to compare the differences between nominal variables. A one-way ANOVA test was used to determine if there was a statistically significant difference between the four categorical groups by testing for differences of means using variance. Crude incidence rates were computed as the total number of study outcomes during the follow-up time divided by person-years at risk. Cox proportional hazards regression was used to compare the risk of events between different INR or aPTT ratio in AF patients after taking DOAC. Statistical significance was defined as a *p* value < 0.05. All analyses were conducted using SAS 9.2 (SAS Institute Inc., Cary, NC, USA).

## 3. Results

Among 3192 AF patients taking rivaroxaban, there were 852 (26.7%), 1260 (39.5%), 437 (13.7%), and 643 (20.1%) patients with an INR < 1.1, 1.1~1.2, 1.3~1.4, and ≥ 1.5, respectively. The mean following-up period was 2.79 ± 0.85, 2.76 ± 1.49, 2.73 ± 1.55, and 2.78 ± 1.55 years for patients having an INR < 1.1, 1.1~1.2, 1.3~1.4, and ≥ 1.5, respectively. (*p* = 0.91). The baseline characteristics of patients with a variety of INR after rivaroxaban are shown in Table 1. In general, those patients with a higher INR after rivaroxaban were taking a higher dose of rivaroxaban, had a higher proportion of “on-label” prescription following either the ROCKET AF or J-ROCKET AF dose criteria, [19,20] a lower HAS-BLED score, a lower prevalence of diabetes mellitus, chronic lung or liver disease, and a higher prevalence of congestive heart failure. In addition, those patients with a higher INR had a higher prevalence of diuretics, beta-blocker, and digoxin use. The CHA_2_DS_2_-VASc score and mean body weight were similar between patients with a variety of INR after taking rivaroxaban (Table 1).

Multivariate analysis demonstrated that the “on-label” prescription following the ROCKET AF or J-ROCKET AF dose criteria and use of digoxin were independent factors associated with an INR ≥ 1.5 after taking rivaroxaban, while patients with a higher HAS-BLED score, the presence of underlying diabetes mellitus, higher hemoglobin and eGFR, and use of diltiazem were independent factors associated with an INR < 1.5 after taking rivaroxaban (Table 2 and Figure 2).

### 3.1. Patients Taking Rivaroxaban with Different INRs

The cumulative incidence curves of thromboembolic and bleeding events in patients with different INR categories are shown in Figure 3. Thromboembolic events appeared to be lower in patients with an INR ≥ 1.5 compared to other groups (*p* = 0.04), while the risk of major bleeding was not significantly different between four groups (*p* = 0.77). Appendix A and Figure 4 show the adjusted hazard ratios (aHRs) and 95% confidential intervals (CIs) of IS/SE and major bleeding of patients with different INRs compared to those with an INR < 1.1. Patients with an INR ≥ 1.5 were associated with a lower risk of IS/SE than those with an INR < 1.1 (aHR: 0.57; [95%CI: 0.37–0.87]; *p* < 0.01). The risk of major bleeding was comparable between patients with an INR ≥ 1.5 and those with an INR < 1.1. The risk of IS/SE and major bleeding did not differ significantly between patients with an INR < 1.1, 1.1~1.2, and 1.3–1.4.

### 3.2. “On-Label Dosing” of Rivaroxaban with Different INRs

We calculated the risk of IS/SE and major bleeding among AF patients taking on-label dosing of rivaroxaban (n = 2203) following either ROCKET AF (20/15 mg once daily) or J-ROCKET AF (15/10 mg once daily) dosage criteria with different INRs. Appendix A and Figure 5 show aHRs and 95%CIs of IS/SE and major bleeding of patients with different INRs compared to those with an INR < 1.1. Consistent with the main analysis, patients with an INR ≥ 1.5 were still associated with a lower risk of IS/SE than those with an INR < 1.1 after taking on-label dosing of rivaroxaban, while the risk of major bleeding was similar.

### 3.3. Patients Taking Dabigatran with Different aPTT Ratios

For 958 patients taking dabigatran, there were 305 (31.8%), 234 (24.4%), 195 (20.4%), and 224 (23.4%) patients having an aPTT ratio < 1.1, 1.1~1.2, 1.3~1.4, and ≥ 1.5, respectively. The mean following-up period was 3.19 ± 1.71, 3.20 ± 1.67, 3.23 ± 1.67, and 2.92 ± 1.74 years for patients having an aPTT ratio < 1.1, 1.1~1.2, 1.3~1.4, and ≥ 1.5, respectively (*p* = 0.18). There were 797 patients (83%) taking low-dose dabigatran 110 mg twice daily. Baseline characteristics of patients with a variety of aPTT ratio after dabigatran are shown in Table 3. In general, those patients with a higher aPTT ratio after being treated with dabigatran were older (*p* < 0.01) and had a higher CHA_2_DS_2_-VASc score (*p* = 0.02).

Patients with an aPTT ratio of 1.1–1.2 and 1.3–1.4 were associated with a lower risk of IS/SE than those with an aPTT ratio < 1.1 after dabigatran (aHR: 0.50; [95%CI: 0.26–0.95]; *p* = 0.04 for an aPTT ratio of 1.1–1.2 vs. < 1.1; aHR: 0.38; [95%CI 0.18–0.81]; *p* = 0.01 for aPTT ratio of 1.3–1.4 vs. < 1.1), while patients with an aPTT ratio ≥1.5 showed a comparable risk of IS/SE to those with an aPTT ratio <1.1. The risk of major bleeding was comparable between patients with a different aPTT ratio (Appendix A and Figure 6). Logistic regression analysis did not demonstrate any factors significantly associated with an elevated aPTT ratio for patients treated with dabigatran.

## 4. Discussion

To our best knowledge, this is the largest study to evaluate the role of common coagulation tests with PT-INR and aPTT ratios in predicting the risks of IS/SE and major bleeding in Asian AF patients taking rivaroxaban and dabigatran in real-world daily practice. The principal findings of this study are as follows: (i) Despite these tests being commonly used by clinicians concerned about over-anticoagulation and bleeding risk, PT-INR or aPTT ratios were not associated with bleeding events for rivaroxaban or dabigatran, respectively; (ii) AF patients with an INR ≥ 1.5 after taking rivaroxaban were associated with a lower risk of IS/SE and a similar risk of major bleeding compared to those with an INR < 1.1, and (iii) For patients treated with dabigatran, a higher aPTT ratio was associated with a lower risk of IS/SE, while the aPTT ratio was not associated with major bleeding.

### 4.1. Factors Associated with the INR Values after Taking Rivaroxaban

In the present study, body weight and CHA_2_DS_2_-VASc score were similar for each subgroup with different INRs after taking rivaroxaban. eGFRs were lower in AF patients with an INR ≥ 1.5 after rivaroxaban compared to other groups. Multivariate analysis also indicated that a lower eGFR was an independent factor associated with a prolongation of INR to ≥ 1.5 after rivaroxaban (Table 2 and Figure 2). Thus, a lower eGFR may increase the plasma concentration of rivaroxaban, and thus potentially increase the INR values for patients taking rivaroxaban. Our study was partially concordant with the previous small study indicating that poor renal function, presence of persistent AF, and absence of prior stroke were associated a higher PT/INR level among AF patients taking rivaroxaban [21]. Interestingly, the use of digoxin was associated with an INR level of ≥ 1.5, while the use of diltiazem was associated with a lower INR. One possible explanation for this observation is that a higher prescription rate of digoxin and a lower prescription of diltiazem were noted among AF patients with congestive heart failure compared to those without in our present study, and previous studies have showed that a prolonged PT-INR level was common observed in heart failure patients possibly due to the liver congestion [22,23]. Therefore, whether the different INR levels for patients receiving digoxin and diltiazem were really due to these medications or were confounded by other factors was unclear. The detailed mechanism(s) behind the associations between these clinical factors and INR levels after rivaroxaban merits further study.

### 4.2. The Role of PT-INR in Patients Taking Rivaroxaban

A few clinical studies have evaluated the clinical outcomes based on coagulation assays among AF patients taking DOACs. Ofek et al. collected the PT-INR data from 218 hospitalized patients treated with rivaroxaban and apixaban. INR was significantly higher in the rivaroxaban group (median 1.7 [1.3–2.5]) than in the apixaban group (median 1.4 [1.2–1.6]); however, there were apparently no other factors affecting INR but the drugs themselves [24]. One small study evaluated the role of PT-INR in only 69 AF patients taking rivaroxaban and found that patients with an INR > 1.5 had a trend for fewer adverse events (bleeding, stroke, and unexpected hospitalizations) than those with an INR < 1.5 [21]. Another study indicated that a PT of ≥ 30 s was associated with a higher risk of bleeding in 199 hospitalized patients receiving rivaroxaban with coagulation tests performed [25]. Recently, one retrospective single center study in Japan indicated that an instance of excessive prolongation in PT or aPTT was associated with a higher bleeding risk but not a lower risk of thromboembolism in 1521 AF patients taking DOACs [26], which is in contrast to our study showing that prolongation of INR was associated with a lower risk of thromboembolism without an increased risk of major bleeding.

There were several reasons which may help to explain the opposite results between our study and the study by Kawabata et al. [26]. First, our study enrolled 3192 patients taking rivaroxaban, which is much larger than the previous study enrolling only 447 patients taking rivaroxaban [26]. Second, a high prevalence of “off-label” over-dose DOACs (6.4%) was noted in the previous study, and the multivariate analysis revealed significant associations of inappropriately high prescription dosages with excessive coagulation time prolongation. In contrast, the prevalence of off-label over-dose rivaroxaban was less than 1% in our study (Table 1). Third, the median PT value was 13.1 s in our present study treated with rivaroxaban (Figure 4), which was numerically lower than that reported by Kawabata et al. (15.3 s). Furthermore, among the study group with an INR level of ≥1.5 (PT value ≥ 16.5 s), the median PT level was 19.7 s [17.4–25.8] in our population, whereas the cut-off value of excessive prolongation was ≥ 23 s in the report by Kawabata et al. In other words, most patients with an INR level of ≥ 1.5 in our present study only showed a “moderate” prolonged PT/INR (e.g., PT < 19.7 s or INR < 1.8) and would not be classified as the excessive prolongation group according to the definition of Kawabata et al. Fourth, a high prevalence of positive anti-phospholipid antibodies (28%) was observed in patients experiencing major bleeding after DOACs in previous study. However, the actual prevalence of antiphospholipid antibody positivity in our multi-center healthcare system or in general population is unknown. Whether common coagulation assays could be useful in the prediction of clinical outcomes in AF patients taking DOACs should be further investigated in prospective studies.

### 4.3. The Role of aPTT in Patients Taking Dabigatran

Different from FXa inhibitors, the aPTT may provide a qualitative assessment of concentration and anticoagulation activity of dabigatran [2]. In the present study, an aPTT ratio between 1.1–1.2 and 1.3–1.4 was associated with a lower risk of IS/TE compared to an aPTT ratio < 1.1, suggesting a better efficacy due to a possibly higher anticoagulation activity. In a prior study which included only 139 patients, the aPTT values were prolonged under dabigatran usage but exhibited a remarkable diversity [27]. During a short follow-up duration (median 120 days), there were no intrinsic major bleeding events while 11 patients had minor hemorrhagic events whose aPTT values ranged widely from 33.7 to 74.3 s. Similar to the findings of our study, the aPTT ratio did not play a prognostic role in the prediction of major bleeding.

### 4.4. Clinical Implications

Although the measurements of PT-INR or aPTT were not performed for DOACs in the randomized trials, these tests were commonly used by clinicians concerned about over-anticoagulation and bleeding risk. Therefore, how to interpret the results of PT-INR and aPTT for patients treated with DOACs is clinically relevant. Based on our data, prolongations of PT-INR and aPTT were not associated with an increased risk of major bleeding in patients treated with rivaroxaban and dabigatran, respectively. Therefore, clinical physicians should not adjust the dosages of rivaroxaban or dabigatran for patients with a prolongation of PT-INR or aPTT once these tests are performed, which, although generally not suggested by current guidelines, are widely practiced, especially in Asian countries. On the contrary, a prolongation of PT-INR to ≥ 1.5 was associated with a lower risk of IS/SE for patients treated with rivaroxaban. In the present study, “on-label” dosing of rivaroxaban was an important independent factor (OR: 1.56; 95%CI: [1.25–1.95]; *p* < 0.01) associated with an INR ≥ 1.5. Since inappropriate prescriptions of low-dose rivaroxaban without following the “label adherent” recommendation may cause more thromboembolic events, [27] appropriate dosages of DOACs and treatment adherence/compliance should be confirmed for patients taking rivaroxaban with an INR < 1.5. We should pay attention to the dosage of rivaroxaban the patients received and try to avoid the “off-label” low-dosing if patients had an INR of < 1.1. For patients who already took “on-label” dosing of rivaroxaban with an INR of < 1.1, we should make sure adherence of patients. For those patients who already took “on-label” dosing following the J-ROCKET AF dose criteria (15/10 mg/day) and had a good drug adherence, a higher-dose of rivaroxaban following the ROCKET-AF dose criteria (20/15 mg/day) may be considered. In addition, any modifiable risk factors and co-medication (e.g., use of diltiazem or digoxin) should be checked carefully once patients are below the target range of INR when taking rivaroxaban. Further large-scaling clinical trials enrolling more AF patients to confirm the findings from the present study are necessary.

### 4.5. Study Limitations

There were several limitations of the present study. First, the precise timing intervals between the last dose of DOACs and blood samplings were difficult to be ascertained due to the retrospective nature of the study. Because the plasma concentrations of rivaroxaban or dabigatran both increase rapidly after intake until reaching the maximum plasma concentration. Second, The aPTT or PT-INR value is speculated to vary according to the time of sampling relative to drug intake. Furthermore, time to peak concentration and the half-life elimination of these drugs are also dependent on multiple factors including gastrointestinal tract absorption (which is important for rivaroxaban), creatinine clearance, age, and other drug interactions. Dabigatran is a twice daily dose, whereas rivaroxaban is a once daily medication. Of note, rivaroxaban was recommended to be taken in the morning (with meal) according to the drug labeling in Taiwan. However, the previous study showed no obvious relationship between the aPTT value and the blood-sampling time for dabigatran [26]. Nevertheless, not considering the sampling time and other factors affecting the blood concentration of DOACs was a major limitation of our present study as well as other retrospective studies [21,26]. The number (n = 224) of patients receiving dabigatran with an aPTT ratio of >1.5 was much smaller than the number (n = 643) of those receiving rivaroxaban with an PT level of >1.5, which makes it hard to make a firm conclusion about whether a further lower risk of ischemic stroke could be observed in those patients receiving dabigatran with an aPTT ratio of >1.5. A further well-designed large prospective study with precise timings of blood sampling is necessary. Second, previous studies indicated that the sensitivity of PT-INR to rivaroxaban and other DOACs was highly dependent on the reagent used. Our multi-center healthcare system used lyophilized human placental thromboplastin, and it is not clear whether the measurements of the PT-INR using other reagents could produce similar findings. Third, the reasons why patients received the coagulation tests after taking DOACs were unclear in the present study. The median duration between the first prescription of DOACs and the measurement of PT-INR or aPTT value was 30 days, while the medical records did not indicate the occurrences of clinical events at the same time. Since more than half of the entire population treated with DOACs (4150 out of 7815 patients, 53%) received the coagulation tests, we believe that the likelihood of selection bias affecting our results was minimal. Fourth, our study was an on-treatment design and did not take into account subsequent changes in medical conditions or activities during follow-up (e.g., newly diagnosed co-morbidities, decreases in eGFR, discontinuation/increase in combination medications, and sequential changes in aPTT or PT values, which are often not obtained during regular visits and may not reflect the actual effects of DOACs on each individual’s aPTT or PT values. Therefore, we did not use sequential aPTT or PT values for analysis. Finally, the present study only enrolled Asian patients; therefore, whether the results can be extrapolated to other non-Asian cohorts remains unclear [28].

## 5. Conclusions

In AF patients, PT-INR or aPTT ratios were not associated with bleeding events for rivaroxaban or dabigatran. Patients taking rivaroxaban with an INR ≥ 1.5 were associated with a lower risk of IS/SE.

## Figures and Tables

**Figure 1 biomedicines-10-03210-f001:**
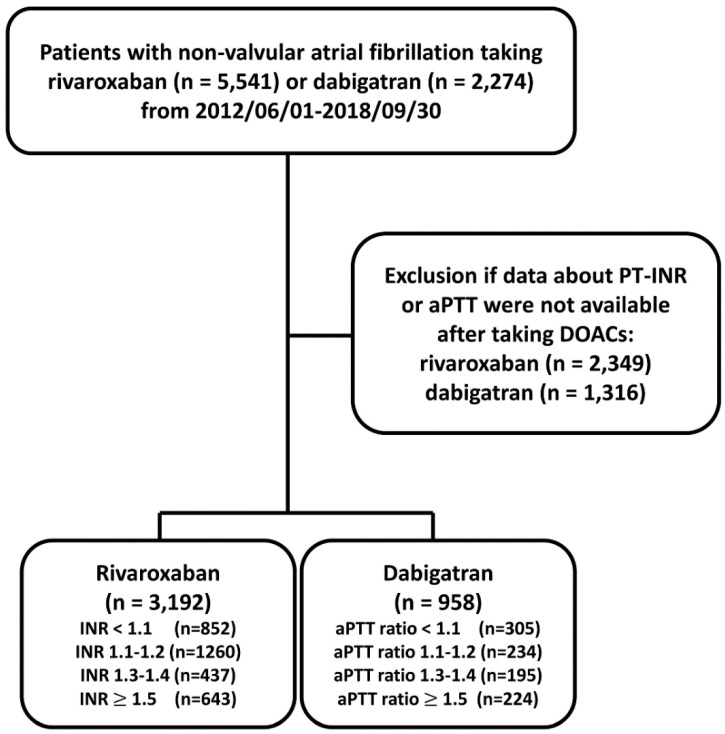
Enrollment of patients with non-valvular atrial fibrillation with baseline prothrombin time-international normalized ratio (PT-INR) available after taking direct oral anticoagulants (DOACs). From 1 June 2012 to 30 September 2018, a total of 5541 and 2274 patients diagnosed with non-valvular AF taking rivaroxaban or dabigatran were identified. The study finally identified a total of 3192 patients taking rivaroxaban and 958 patients taking dabigatran with PT-INR and aPTT available, respectively. aPTT = activated partial thromboplastin time; DOACs = direct oral anticoagulants; PT-INR = prothrombin time-international normalized ratio.

**Figure 2 biomedicines-10-03210-f002:**
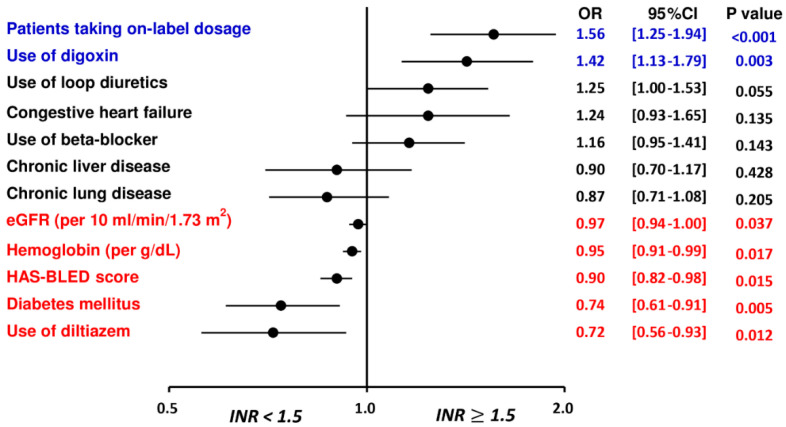
Factors associated with an INR ≥ 1.5 for patients treated with rivaroxaban. The multivariate analysis indicated that the “on-label” prescription following the ROCKET AF or J-ROCKET AF dose criteria and use of digoxin were independent factors associated with an INR ≥ 1.5 after taking rivaroxaban, while patients with a higher HAS-BLED score, the presence of underlying diabetes mellitus, higher hemoglobin and eGFR, and use of diltiazem were independent factors associated with an INR < 1.5 after taking rivaroxaban. CHA_2_DS_2_-VASc = congestive heart failure, hypertension, age 75 years or older, diabetes mellitus, previous stroke/transient ischemic attack, vascular disease, age 65 to 74 years, female; CI = confidence interval; eGFR = estimated glomerular filtration rate; HAS-BLED = hypertension, abnormal renal or liver function, stroke, bleeding history, labile INR, age 65 years or older, and antiplatelet drug or alcohol use; INR = international normalized ratio; NVAF = non-valvular atrial fibrillation; OR = odds ratio.

**Figure 3 biomedicines-10-03210-f003:**
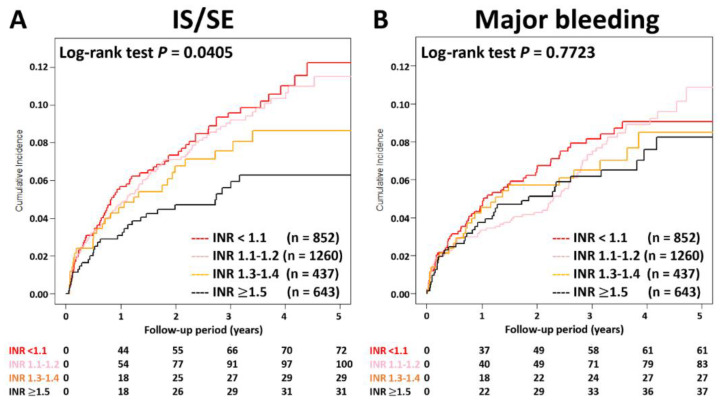
Cumulative incidence curves of IS/SE and major bleeding for AF patients treated with rivaroxan with different INR categories. Thromboembolic events appeared to be lower in patients with an INR ≥ 1.5 compared to other groups (*p* = 0.04), while the risk of major bleeding was not significantly different between four groups (*p* = 0.77). AF = atrial fibrillation; INR = international normalized ratio; IS/SE = ischemic stroke/systemic embolism.

**Figure 4 biomedicines-10-03210-f004:**
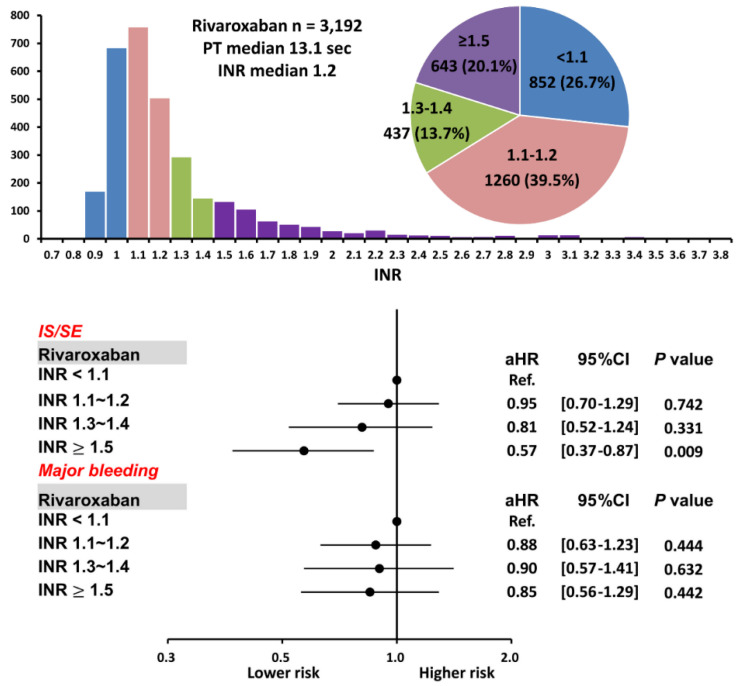
Risks of IS/SE and major bleeding for 3192 AF patients treated with rivaroxaban with different INR. Among 3192 patients treated with rivaroxaban, there were 643 (20.1%) patients having an INR ≥ 1.5. Compared to patients with an INR < 1.1, patients with an INR ≥ 1.5 were associated with a lower risk of IS/SE and were not associated with a higher risk of major bleeding. aHR = adjusted hazard ratio; CI = confidential interval; INR = international normalized ratio; IS/SE = ischemic stroke/systemic embolism; PT = prothrombin time.

**Figure 5 biomedicines-10-03210-f005:**
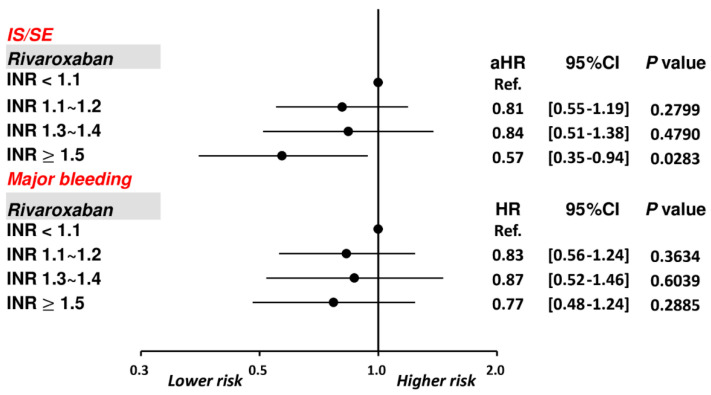
Risks of IS/SE and major bleeding for 3192 AF patients treated with on-label dosing of rivaroxaban with different INR levels. Among 2203 patients treated with rivaroxaban, compared to patients with an INR < 1.1, patients with an INR ≥ 1.5 were associated with a lower risk of IS/SE and were not associated with a higher risk of major bleeding. aHR = adjusted hazard ratio; CI = confidential interval; INR = international normalized ratio; IS/SE = ischemic stroke/systemic embolism; PT = prothrombin time.

**Figure 6 biomedicines-10-03210-f006:**
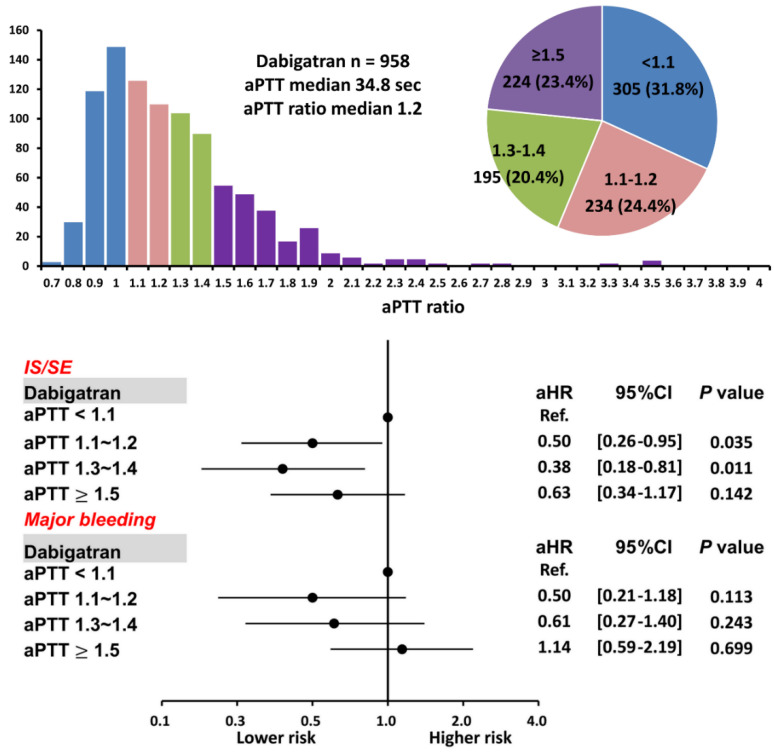
Risks of IS/SE and major bleeding for 958 AF patients treated with dabigatran with different aPTT ratio levels. For patients treated with dabigatran, a higher aPTT ratio of 1.1–1.2 and 1.3–1.4 were associated with a lower risk of IS/SE than those with an aPTT ratio <1.1. A prolonged aPTT ratio was not associated with a higher risk of major bleeding. aPTT = activated partial thromboplastin time; CI = confidential interval; aHR = adjusted hazard ratio; IS/SE = ischemic stroke/systemic embolism; NVAF = non-valvular atrial fibrillation.

**Table 1 biomedicines-10-03210-t001:** Baseline characteristics of AF patients treated with rivaroxaban with different INR levels.

	Rivaroxaban				
	INR < 1.1 (*n* = 852)	INR = 1.1~1.2 (*n* = 1260)	INR = 1.3~1.4 (*n* = 437)	INR ≥ 1.5 (*n* = 643)	*p* Value (ANOVA)
**Coagulation test**					
**PT (s), median value [IQR]**	10.7 [10.4–11.1]	12.4 [11.9–13.1] *	14.6 [14.0–15.3] *^,†^	19.7 [17.4–25.8] ^v,^*^,†^	<0.01
**INR, median value**	1.01 [1.00–1.05]	1.15 [1.10–1.20] *	1.32 [1.30–1.40] *^,†^	1.80 [1.60–2.30] ^v,^*^,†^	<0.01
**Rivaroxaban Dosage**					
**Rivaroxaban 10 mg once daily, n (%)**	391 (45.89%)	538 (42.7%)	149 (34.1%) *^,†^	199 (30.95%) *^,†^	<0.01
**Rivaroxaban 15 mg once daily, n (%)**	410 (48.12%)	635 (50.4%)	242 (55.38%) *	370 (57.54%) *^,†^	<0.01
**Rivaroxaban 20 mg once daily, n (%)**	51 (5.99%)	87 (6.9%)	46 (10.53%) *^,†^	74 (11.51%) *^,†^	<0.01
**Patients following ROCKET AF dose criteria, n (%)**	104 (12.21%)	179 (14.21%)	79 (18.08%) *	134 (20.84%) *^,†^	<0.01
**Patients following with J-ROCKET AF dose criteria, n (%)**	448 (52.58%)	666 (52.86%)	244 (55.84%)	349 (54.28%)	0.66
**Patients following ROCKET AF or J-ROCKET AF dose criteria, n (%)**	552 (64.79%)	845 (67.06%)	323 (73.91%) *^,†^	483 (75.12%) *^,†^	<0.01
**Patients taking off-label underdose, n (%)**	288 (33.8%)	396 (31.43%)	101 (23.11%) *^,†^	140 (21.77%) *^,†^	<0.01
**Patients taking off-label overdose, n (%)**	3 (0.35%)	6 (0.48%)	4 (0.92%)	5 (0.78%)	0.50
**Baseline characteristics**					
**Age, yrs**	74.03 ± 10.12	75.24 ± 10.2 *	75.19 ± 10.81	74.36 ± 10.67	0.03
**Female, n (%)**	393 (46.13%)	496 (39.37%) *	172 (39.36%) *	286 (44.48%) ^†^	0.01
**CHA_2_DS_2_-VASc**	3.67 ± 1.63	3.63 ± 1.63	3.61 ± 1.6	3.6 ± 1.59	0.75
**HAS-BLED**	2.91 ± 1.23	2.9 ± 1.23	2.76 ± 1.16 *^,†^	2.66 ± 1.19 *^,†,v^	<0.01
**Past medical history, n (%)**					
**Chronic lung disease**	271 (31.81%)	440 (34.92%)	135 (30.89%)	184 (28.62%) ^†^	0.04
**Chronic liver disease**	199 (23.36%)	303 (24.05%)	82 (18.76%) ^†^	116 (18.04%) *^,†^	<0.01
**Congestive heart failure**	92 (10.8%)	144 (11.43%)	60 (13.73%)	104 (16.17%) *^,†^	<0.01
**Hypertension**	668 (78.4%)	953 (75.63%)	335 (76.66%)	484 (75.27%)	0.43
**Hyperlipidemia**	385 (45.19%)	565 (44.84%)	187 (42.79%)	258 (40.12%) ^†^	0.18
**Diabetes mellitus**	330 (38.73%)	462 (36.67%)	164 (37.53%)	199 (30.95%) *^,†,v^	0.01
**Previous stroke**	169 (19.84%)	248 (19.68%)	72 (16.48%)	135 (21%)	0.32
**Previous TIA**	14 (1.64%)	31 (2.46%)	14 (3.2%)	11 (1.71%)	0.22
**Ischemic heart disease**	103 (12.09%)	153 (12.14%)	53 (12.13%)	70 (10.89%)	0.86
**Gout**	139 (16.31%)	230 (18.25%)	88 (20.14%)	108 (16.8%)	0.32
**Peripheral artery disease**	3 (0.35%)	3 (0.24%)	1 (0.23%)	0 (0%)	0.55
**Malignancy**	132 (15.49%)	223 (17.7%)	89 (20.37%) *	108 (16.8%)	0.17
**Baseline laboratory data**					
**Hemoglobin, g/dL**	12.87 ± 2.12	12.7 ± 2.21 *	12.31 ± 2.41 *^,†^	12.51 ± 2.37 *	<0.01
**Platelet, × 1000/Ul**	204.11 ± 68.11	197.79 ± 70.32	187.88 ± 78.57 *^,†^	195.89 ± 74.53	0.01
**eGFR, mL/min/1.73 m^2^**	75.19 ± 31.48	75.97 ± 31.89	75.44 ± 35.1	71.99 ± 30.86 ^†,v^	0.08
**ALT, U/L**	27.79 ± 22.9	27.18 ± 20.16	27.06 ± 21.3	27.31 ± 23.86	0.93
**Height, cm**	160.16 ± 8.38	160.60 ± 8.89	160.78 ± 9.10	160.40 ± 9.46	0.71
**Body weight, kg**	64.35 ± 12.90	65.03 ± 13.95	64.30 ± 13.76	63.60 ± 13.33	0.27
**Baseline medications, n (%)**					
**Use of NSAIDs**	131 (15.38%)	187 (14.84%)	66 (15.1%)	97 (15.09%)	0.99
**Use of PPI**	140 (16.43%)	159 (12.62%) *	56 (12.81%)	81 (12.6%) *	0.06
**Use of ACEI/ARB**	465 (54.58%)	720 (57.14%)	247 (56.52%)	385 (59.88%) *	0.24
**Use of loop diuretics**	230 (27%)	456 (36.19%) *	157 (35.93%) *	251 (39.04%) *	<0.01
**Use of amiodarone**	206 (24.18%)	264 (20.95%)	74 (16.93%) *	125 (19.44%) *	0.01
**Use of dronedarone**	24 (2.82%)	44 (3.49%)	19 (4.35%)	23 (3.58%)	0.55
**Use of quinidine**	0 (0%)	2 (0.16%)	1 (0.23%)	2 (0.31%)	0.48
**Use of beta-blocker**	462 (54.23%)	716 (56.83%)	250 (57.21%)	395 (61.43%) *	0.05
**Use of diltiazem**	161 (18.9%)	278 (22.06%)	94 (21.51%)	110 (17.11%)	0.05
**Use of verapamil**	40 (4.69%)	65 (5.16%)	21 (4.81%)	23 (3.58%)	0.49
**Use of digoxin**	136 (15.96%)	235 (18.65%)	92 (21.05%) *	164 (25.51%) *^,†^	<0.01
**Use of statin**	280 (32.86%)	437 (34.68%)	150 (34.32%)	212 (32.97%)	0.80
**Use of Azi-/Clari-/Erythromycin**	15 (1.76%)	36 (2.86%)	6 (1.37%)	9 (1.4%) ^†^	0.08
**Use of Itraconzaole**	1 (0.12%)	1 (0.08%)	0 (0%)	0 (0%)	0.30
**Use of cyclosporin**	3 (0.35%)	1 (0.08%)	2 (0.46%)	1 (0.16%)	0.75

ACEI = angiotensin-converting-enzyme inhibitor; AF = atrial fibrillation; ALT = alanine aminotransferase; ARB = angiotensin II receptor antagonists; CHA_2_DS_2_-VASc = congestive heart failure, hypertension, age 75 years or older, diabetes mellitus, previous stroke/transient ischemic attack, vascular disease, age 65 to 74 years, female; eGFR = estimated Glomerular filtration rate; HAS-BLED = hypertension, abnormal renal or liver function, stroke, bleeding history, labile INR, age 65 years or older, and antiplatelet drug or alcohol use; INR = international normalized ratio; IQR = interquartile range; NSAIDs = non-steroidal anti-inflammatory drugs; PPI = proton pump inhibitor; TIA = transient ischemic attack. * *p* < 0.05 vs. INR < 1.1. ^†^
*p* < 0.05 vs. INR 1.1~1.2. ^v^
*p* < 0.05 vs. INR 1.3~1.4.

**Table 2 biomedicines-10-03210-t002:** Factors associated with an INR ≥ 1.5 after rivaroxaban.

	Univariate Odds Ratio (OR)		Multivariate Odds Ratio (OR)	
	OR (95% CI)	*p* Value	OR (95% CI)	*p* Value
**Patients treated with on-label dosing** **of rivaroxaban**	**1.45 (1.19–1.77)**	**<0.01**	**1.56 (1.25–1.94)**	**<0.01**
**Age**	1.00 (0.99–1.00)	0.31		
**Female**	1.12 (0.94–1.34)	0.19		
**Body weight**	0.99 (0.99–1.00)	0.11		
**CHA_2_DS_2_-VASc score**	0.97 (0.92–1.03)	0.32		
**HAS-BLED score**	**0.86 (0.80–0.93)**	**<0.01**	**0.90 (0.82–0.98)**	**0.02**
**Chronic lung disease**	**0.81 (0.67–0.98)**	**0.03**	0.87 (0.71–1.08)	0.21
**Chronic liver disease**	**0.74 (0.59–0.92)**	**<0.01**	0.90 (0.70–1.17)	0.43
**Congestive heart failure**	**1.47 (1.15–1.87)**	**<0.01**	1.24 (0.93–1.65)	0.14
**Diabetes mellitus**	**0.75 (0.62–0.90)**	**<0.01**	**0.74 (0.61–0.91)**	**<0.01**
**Hemoglobin, per g/dL**	**0.95 (0.91–0.99)**	**0.02**	**0.95 (0.91–0.99)**	**0.02**
**Platelet, per 10,000/uL**	1.00 (1.00–1.00)	0.53		
**eGFR, per 10 mL/min/1.73 m^2^**	**0.96 (0.93–0.99)**	**<0.01**	**0.97 (0.94–1.00)**	**0.04**
**Use of loop diuretics**	**1.30 (1.08–1.55)**	**<0.01**	1.24 (1.00–1.53)	0.06
**Use of amiodarone**	0.89 (0.72–1.11)	0.29		
**Use of beta-blocker**	**1.25 (1.05–1.49)**	**0.01**	1.16 (0.95–1.41)	0.14
**Use of diltiazem**	**0.78 (0.62–0.98)**	**0.03**	**0.72 (0.56–0.93)**	**0.01**
**Use of digoxin**	**1.54 (1.26–1.89)**	**<0.01**	**1.42 (1.13–1.79)**	**<0.01**

CI = confidential interval; OR = odds ratio; other abbreviations as in Table 1.

**Table 3 biomedicines-10-03210-t003:** Baseline characteristics of AF patients treated with dabigatran with different aPTT ratio levels.

	Dabigatran				
	aPTT Ratio < 1.1 (*n* = 305)	aPTT Ratio = 1.1~1.2 (*n* = 234)	aPTT Ratio = 1.3~1.4 (*n* = 195)	aPTT Ratio ≥ 1.5 (*n* = 224)	*p* Value (ANOVA)
**Coagulation test**					
**aPPT (s), median value [IQR]**	27.9 [26.4–29.1]	33.4 [32.2–34.8] *	38.9 [37.5–40.5] *^,†^	48.1 [44.8–54.1] ^v,^*^,†^	<0.01
**aPTT ratio, median value**	1.00 [0.94–1.04]	1.19 [1.15–1.24] *	1.39 [1.34–1.45] *^,†^	1.72 [1.60–1.93] ^v,^*^,†^	<0.01
**Dabigatran Dosage**					
**Dabigatran 110 mg twice daily, n (%)**	254 (83%)	197 (84%)	165 (85%)	181 (81%)	0.72
**Dabigatran 150 mg twice daily, n (%)**	51 (17%)	37 (16%)	30 (15%)	43 (19%)	0.72
**Baseline characteristics**					
**Age, yrs**	72.39 ± 10.19	72.02 ± 9.82	74.20 ± 10.30	75.03 ± 10.13 *^,†^	<0.01
**Female, n (%)**	113 (37%)	76 (32%)	62 (32%)	91 (41%)	0.08
**CHA_2_DS_2_-VASc**	3.10 ± 1.32	3.09 ± 1.43	3.35 ± 1.52	3.39 ± 1.40	0.02
**HAS-BLED**	2.83 ± 1.16	2.73 ± 1.17	2.86 ± 1.26	2.88 ± 1.09	0.56
**Past medical history, n (%)**					
**Chronic lung disease**	91 (30%)	68 (29%)	56 (29%)	72 (32%)	0.86
**Chronic liver disease**	54 (18%)	53 (23%)	40 (21%)	35 (16%)	0.23
**Congestive heart failure**	19 (6%)	13 (6%)	17 (9%)	18 (8%)	0.52
**Hypertension**	238 (78%)	184 (79%)	142 (73%)	167 (75%)	0.41
**Hyperlipidemia**	138 (45%)	112 (48%)	88 (45%)	99 (44%)	0.88
**Diabetes mellitus**	120 (39%)	75 (32%)	71 (36%)	89 (40%)	0.27
**Previous stroke**	76 (25%)	64 (27%)	64 (33%)	68 (30%)	0.24
**Previous TIA**	7 (2%)	11 (4%)	11 (6%)	10 (4%)	0.28
**Ischemic heart disease**	25 (8%)	13 (10%)	13 (7%)	24 (11%)	0.43
**Gout**	47 (15%)	38 (12%)	38 (19%)	43 (19%)	0.13
**Peripheral artery disease**	1 (0%)	1 (0%)	1 (1%)	0 (0%)	0.57
**Malignancy**	53 (17%)	28 (15%)	28 (14%)	38 (17%)	0.72
**Baseline laboratory data**					
**Hemoglobin, g/dL**	13.25 ± 2.20	13.71 ± 1.91	13.51 ± 1.99	12.96 ± 2.15 ^†^	<0.01
**Platelet, × 1000/Ul**	202.18 ± 65.80	203.17 ± 58.84	207.76 ± 77.75	207.84 ± 81.07	0.76
**eGFR, ml/min/1.73 m^2^**	79.28 ± 31.13	80.66 ± 24.10	76.69 ± 24.86	74.90 ± 29.79	0.12
**ALT, U/L**	25.99 ± 29.96	26.73 ± 20.28	26.17 ± 19.41	24.73 ± 17.63	0.75
**Height, cm**	161.56 ± 9.38	162.51 ± 8.11	162.28 ± 8.30	159.60 ± 8.87 ^†^	0.01
**Body weight, kg**	65.61 ± 13.22	68.32 ± 15.69	66.41 ± 11.14	64.44 ± 13.69	0.06
**Baseline medications, n (%)**					
**Use of NSAIDs**	47 (15%)	29 (12%)	33 (17%)	35 (16%)	0.59
**Use of PPI**	38 (12%)	19 (8%)	18 (9%)	22 (10%)	0.38
**Use of ACEI/ARB**	161 (53%)	122 (52%)	110 (56%)	120 (54%)	0.83
**Use of loop diuretics**	70 (23%)	43 (18%)	43 (22%)	54 (24%)	0.47
**Use of amiodarone**	65 (21%)	56 (24%)	33 (17%)	42 (19%)	0.29
**Use of dronedarone**	4 (1%)	4 (2%)	0 (0%)	1 (0%)	0.22
**Use of quinidine**	0 (0%)	0 (0%)	0 (0%)	0 (0%)	1.00
**Use of beta-blocker**	170 (56%)	120 (51%)	97 (51%)	127 (57%)	0.38
**Use of diltiazem**	54 (18%)	39 (17%)	31 (16%)	48 (21%)	0.45
**Use of verapamil**	8 (3%)	6 (3%)	8 (4%)	11 (5%)	0.41
**Use of digoxin**	34 (11%)	40 (17%)	30 (15%) *	36 (16%)	0.21
**Use of statin**	108 (35%)	91 (39%)	72 (37%)	84 (38%)	0.87
**Use of Azi-/Clari-/Erythromycin**	6 (2%)	5 (2%)	2 (1%)	4 (2%)	0.83
**Use of Itraconzaole**	0 (0%)	0 (0%)	0 (0%)	0 (0%)	1.00
**Use of cyclosporin**	1 (0%)	0 (0%)	0 (0%)	0 (0%)	0.54

Abbreviations as in Table 1. * *p* < 0.05 vs. aPTT ratio < 1.1. ^†^
*p* < 0.05 vs. aPTT ratio 1.1~1.2. ^v^
*p* < 0.05 vs. aPTT ratio 1.3~1.4.

## Data Availability

Not applicable.

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
