# Peer review of "Prothrombin Time-International Normalized Ratio Predicts the Outcome of Atrial Fibrillation Patients Taking Rivaroxaban"

_biomedicines, 2022, doi:10.3390/biomedicines10123210_

Round 1
Reviewer 1 Report
The authors present interesting study aiming to assess the usefulness of routine blood tests such as INR in AF patients treated with rivaroxaban and aPTT in AF patients treated with dabigatran. They have found that INR > 1.5 in patients with rivaroxaban is associated with lower risk of ischemic stroke without increased risk of bleeding, while aPTT does not have any significant value for prognosis in patients with dabigatran. To my opinion this study is interesting from the clinical point as these tests are widely used in clinical practice and it is not clear wether it is necessary to correct the dosage of DOACS in case of increased INR or aPTT. The authors findings tell that the correction is not necessary.
I have several comments.
1) The authors tested several levels of INR and aPTT (< 1.1, 1.1-1.2, 1.3-1.4, > 1.5).
Could the authors explain why they chose such four levels?
In the section introduction the authors mentioned the level of 1.5 as a cut-off threshold to determine the effective anticoagulation in patients treated with warfarin and heparin. I agree that aPTT 1.5-2.5 is therapeutic range for heparin, but for warfarine the therapeutic range is 2-3. The publication (ref 3) to which the author apply is focused on laboratory monitoring of drug levels (DOACs) but not on the INR level.
2) Statistics. The authors used ANOVA test (mentioned in the Table), but there is no information about this test in the text.
3) The authors have found that aPTT 1.1-1.2 and 1.3-1.4 were associated with lower risk of IS and concluded in the discussion that higher level of aPTT is associated with lower risk of IS. However, the level >1.5 is not associated with IS. How could the authors explain this?
4) The figure legends is mixed up. Please, check.
Author Response
Comments for Authors:
Referee: 1
The authors present an interesting study aiming to assess the usefulness of routine blood tests such as INR in AF patients treated with rivaroxaban and aPTT in AF patients treated with dabigatran. They have found that INR > 1.5 in patients with rivaroxaban is associated with a lower risk of ischemic stroke without increased risk of bleeding, while aPTT does not have any significant value for prognosis in patients with dabigatran. To my opinion, this study is interesting from the clinical point as these tests are widely used in clinical practice and it is not clear whether it is necessary to correct the dosage of DOACS in case of increased INR or aPTT. The authors' findings tell that the correction is not necessary.
Response: Thank you. We hope that the revisions have clarified the issues and the overall quality has been improved.
I have several comments.
- The authors tested several levels of INR and aPTT (< 1.1, 1.1-1.2, 1.3-1.4, > 1.5). Could the authors explain why they chose such four levels?
Response: Thank you. We followed the study by Huang et al., (Ref. 1) who classified their study patients treated with rivaroxaban into three groups: INR < 1.1, INR 1.1-1.4, and INR > 1.5. We further separate our patient group with INR 1.1-1.5 into two groups with INR 1.1-1.2 and INR 1.3-1.4 due to the relatively large study population in our present study.
Reference
Huang JH, Lin YK, Chung CC, Hsieh MH, Chiu WC, Chen YJ. Factors That Determine the Prothrombin Time in Patients With Atrial Fibrillation Receiving Rivaroxaban. Clin Appl Thromb Hemost. 2018 Dec; 24(9 Suppl): 188S–193S.
- In the section introduction, the authors mentioned the level of 1.5 as a cut-off threshold to determine the effective anticoagulation in patients treated with warfarin and heparin. I agree that aPTT 1.5-2.5 is the therapeutic range for heparin, but for warfarin, the therapeutic range is 2-3. The publication (ref 3) to which the author apply is focused on laboratory monitoring of drug levels (DOACs) but not on the INR level.
Response: Thank you. Both the European Society of Cardiology (ESC) and the American Heart Association (AHA) recommends a target INR range of 2.0 to 3.0 for the prevention of thromboembolism in patients with NVAF, where the lowest risk of thromboembolism and bleeding can be only achieved in such a narrow therapeutic range. However, several studies indicated that Asians are more sensitive to warfarin and vulnerable to warfarin-related bleeding than Non-Asians (Refs. 1 and 2). The meta-analysis indicated that low-intensity warfarin therapy (INR target of 1.5–2.5) can achieve reduced hemorrhage without increasing thromboembolism for Asian patients with NVAF taking warfarin (Refs. 3 to 6). Therefore, that is why we aimed to investigate whether there was an association between PT-INR (e.g. also using the cut-off value of 1.50), and the risks of ischemic stroke/systemic embolism (IS/SE) and major bleeding among AF patients taking rivaroxaban.
We agreed with the Reviewer’s concern that the publication (ref 3) to we apply is focused on laboratory monitoring of drug levels (DOACs) but not on the INR level. We have corrected it in the revised manuscript.
References
- Shen AY, Yao JF, Brar SS, Jorgensen MB, Chen W. Racial/ethnic differences in the risk of intracranial hemorrhage among patients with atrial fibrillation. J Am Coll Cardiol. 2007; 50(4):309–315.
- van Asch CJ, Luitse MJ, Rinkel GJ, van der Tweel I, Algra A, Klijn CJ. Incidence, case fatality, and functional outcome of intracerebral hemorrhage over time, according to age, sex, and ethnic origin: a systematic review and meta-analysis. Lancet Neurol. 2010; 9(2):167–176.
- You JH, Chan FW, Wong RS, Cheng G. Is INR between 2.0 and 3.0 the optimal level for Chinese patients on warfarin therapy for moderate-intensity anticoagulation? Br J Clin Pharmacol. 2005; 59(5):582–587.
- Inoue H, Okumura K, Atarashi H, Yamashita T, Origasa H, Kumagai N, et al. Target international normalized ratio values for preventing thromboembolic and hemorrhagic events in Japanese patients with non-valvular atrial fibrillation: results of the J-RHYTHM Registry. Circ J. 2013; 77(9):2264–2270.
- Cheung CM, Tsoi TH, Huang CY. The lowest effective intensity of prophylactic anticoagulation for patients with atrial fibrillation. Cerebrovasc Dis. 2005; 20(2):114–119.
- Liu T, Hui J, Hou YY, Zou Y, Jiang WP, Yang XJ, et al. Meta-Analysis of Efficacy and Safety of Low-Intensity Warfarin Therapy for East Asian Patients With Nonvalvular Atrial Fibrillation. Am J Cardiol. 2017; 120(9):1562–1567. pmid:28847595
Page 5, Line 14, Introduction: “Because the INR or aPTT ratio value of 1.5 is commonly used as a cut-off threshold to determine the effective anticoagulant activity in patients treated with warfarin (INR value of 1.5 for Asians specifically) or heparin,(14,15) we aimed to investigate whether there was an association between PT-INR or aPTT ratio (e.g. also using the cut-off value of 1.50), and the risks of ischemic stroke/systemic embolism (IS/SE) and major bleeding among AF patients taking rivaroxaban or dabigatran, respectively.
- Liu T, Hui J, Hou YY, Zou Y, Jiang WP, Yang XJ, Wang XH. Meta-Analysis of Efficacy and Safety of Low-Intensity Warfarin Therapy for East Asian Patients With Nonvalvular Atrial Fibrillation. Am J Cardiol. 2017 Nov 1;120(9):1562-1567.
- Eikelboom JW, Jack Hirsh J. Monitoring unfractionated heparin with the aPTT: time for a fresh look. Thromb Haemost. 2006 Nov;96(5):547-52.
- The authors used the ANOVA test (mentioned in the Table), but there is no information about this test in the text.
Response: Thank you. We have added it to the revised manuscript, as followed:
Page 10, Line 15, Methods: “A one-way ANOVA test was used to determine if there is a statistically significant difference between the four categorical groups by testing for differences of means using a variance.“
- The authors have found that aPTT 1.1-1.2 and 1.3-1.4 were associated with a lower risk of IS and concluded in the discussion that a higher level of aPTT is associated with a lower risk of IS. However, a level >1.5 is not associated with IS. How could the authors explain this?
Response: Thank you. The previous review indicated that although the aPTT was often prolonged under the use of dabigatran, the degree of aPTT prolongation correlated poorly with drug concentration, especially at higher aPTT values. Conversely, rivaroxaban generally prolonged the PT in a concentration-dependent manner, but the correlation was generally weak and became weaker with increasing concentrations (> 50-100 ng/mL). The above conclusion may help explain why a further elevated aPTT > 1.5 is not further associated with a lower risk of ischemic stroke in those patients treated with dabigatran. Another explanation is the number (n = 224) of those treated with dabigatran with an aPTT of > 1.5 was much smaller than the number (n = 643) of those treated with rivaroxaban with a PT level of > 1.5, which make us hard to make a firm conclusion that whether a further lower risk of ischemic stroke was observed in those patients treated with dabigatran with an aPTT level of > 1.5. We fully acknowledge the limitations of our retrospective and observational cohort and have addressed these issues in the revised manuscript.
Page 22, Line 9, Limitations: “The number (n = 224) of patients receiving dabigatran with an aPTT ratio of > 1.5 was much smaller than the number (n = 643) of those receiving rivaroxaban with a PT level of > 1.5, which make us hard to make a firm conclusion whether a further lower risk of ischemic stroke could be observed in those patients receiving dabigatran with an aPTT ratio of > 1.5.“
The figure legends are mixed up. Please, check.
Response: Thank you. We have corrected it in the revised manuscript.
Reviewer 2 Report
I’ve read with attention the paper of Chao et al. that is potentially of interest. The background and aim of the study have been clearly defined. The methodology applied is overall correct, the results are reliable and adequately discussed. My only concern is related to the clinical impact of the reported results. In fact the authors stress the concept that in Asian countries coagulation parameters are monitored during DOAC treatment, even if this is not indicated by the guidelines nor justified by the DOAC mechanism of action. So, the whole study seems to start from a wrong behaviour (nor references in the text). On the other side, it is not clear what the physicians should do (in the authors opinion) if a low INR is found. This part of the discussion should be strongly improved and supported by literature data.
Author Response
Comments for Authors:
Referee: 2
I’ve read with attention the paper of Chao et al. that is potential of interest. The background and aim of the study have been clearly defined. The methodology applied is overall correct, and the results are reliable and adequately discussed. My only concern is related to the clinical impact of the reported results. In fact, the authors stress the concept that in Asian countries coagulation parameters are monitored during DOAC treatment, even if this is not indicated by the guidelines nor justified by the DOAC mechanism of action. So, the whole study seems to start from a wrong behavior (no references in the text). On the other side, it is not clear what the physicians should do (in the authors' opinion) if a low INR is found. This part of the discussion should be strongly improved and supported by literature data.
Response: Thank you. We hope that the revisions have clarified the issues and the overall quality has been improved. We have corrected it in the revised manuscript, as followed:
Page 20, Line 4, Clinical Implications: “
Although the measurements of PT-INR or aPTT were not performed for DOACs in the randomized trials, these tests were commonly used by clinicians concerned about over-anticoagulation and bleeding risk. Therefore, how to interpret the results of PT-INR and aPTT for patients treated with DOACs is clinically relevant. Based on our data, prolongations of PT-INR and aPTT were not associated with an increased risk of major bleeding in patients treated with rivaroxaban and dabigatran, respectively. Therefore, clinical physicians should not adjust the dosages of rivaroxaban or dabigatran for patients with prolongation of PT-INR or aPTT once these tests were performed although generally not being suggested by current guidelines, but widely practiced especially in Asian countries. On the contrary, a prolongation of PT-INR to > 1.5 was associated with a lower risk of IS/SE for patients treated with rivaroxaban. In the present study, “on-label” dosing of rivaroxaban was an important independent factor (OR: 1.56; 95%CI: [1.25-1.95]; P < 0.01) associated with an INR ≥ 1.5. Since inappropriate prescriptions of low-dose rivaroxaban without following the “label adherent” recommendation may cause more thromboembolic events,(26) appropriate dosages of DOACs and treatment adherence/compliance should be confirmed for patients taking rivaroxaban with an INR < 1.5. We should pay attention to the dosage of rivaroxaban the patients received and try to avoid the “off-label” low-dosing if patients had an INR of < 1.1. For patients who already took “on-label” dosing of rivaroxaban with an INR of < 1.1, we should make sure adherence of patients. For those patients who already took “on-label” dosing following the J-ROCKET AF dose criteria (15/10 mg/day) and had good drug adherence, a higher dose of rivaroxaban following the ROCKET-AF dose criteria (20/15 mg/day) may be considered. Also, any modifiable risk factors and co-medication (e.g. use of diltiazem or digoxin) should be checked carefully once patients are below the target range of INR when taking rivaroxaban. Further large-scaling clinical trials enrolling more AF patients to confirm the findings from the present study are necessary.
Reviewer 3 Report
The authors aimed to investigate whether there is an association between prothrombin time and international normalized ratio (PT-INR) or activated partial thromboplastin time (aPTT) ratio, and the risks of ischemic stroke / systemic embolism (IS / SE) and major bleeding among AF patients taking rivaroxaban or dabigatran. This multi-center cohort study in Taiwan included 3,192 AF patients taking rivaroxaban and 958 patients taking dabigatran for stroke prevention where data about PT-INR and aPTT were available.
The authors obtained the following results: for patients treated with rivaroxaban, a higher INR level was not associated with a higher risk of major bleeding compared to an INR level <1.1. The risk of IS / SE was lower for patients having an INR ≥ 1.5 compared to those with an INR <1.1 (aHR: 0.57; [95% CI: 0.37-0.87]; P = 0.01). On-label dosing of rivaroxaban and use of digoxin were independent factors associated with an INR ≥ 1.5 after taking rivaroxaban. For patients taking dabigatran, a higher aPTT ratio was not associated with a higher risk of major bleeding. The risk of IS / SE was lower for patients having an aPTT ratio of 1.1-1.2 and 1.3-1.4 than those with an aPTT ratio <1.1.
This study is well written. Nevertheless, in the introduction or in the discussion, it is worth referring to current studies relating to stroke associated with atrial fibrillation in patients treated with VKAs and DOACs, e.g. Powers et al, Wańkowicz et al.
Author Response
Comments for Authors:
Referee: 3
The authors aimed to investigate whether there is an association between prothrombin time and international normalized ratio (PT-INR) or activated partial thromboplastin time (aPTT) ratio, and the risks of ischemic stroke / systemic embolism (IS / SE) and major bleeding among AF patients taking rivaroxaban or dabigatran. This multi-center cohort study in Taiwan included 3,192 AF patients taking rivaroxaban and 958 patients taking dabigatran for stroke prevention where data about PT-INR and aPTT were available.
The authors obtained the following results: for patients treated with rivaroxaban, a higher INR level was not associated with a higher risk of major bleeding compared to an INR level <1.1. The risk of IS / SE was lower for patients having an INR ≥ 1.5 compared to those with an INR <1.1 (aHR: 0.57; [95% CI: 0.37-0.87]; P = 0.01). On-label dosing of rivaroxaban and use of digoxin were independent factors associated with an INR ≥ 1.5 after taking rivaroxaban. For patients taking dabigatran, a higher aPTT ratio was not associated with a higher risk of major bleeding. The risk of IS / SE was lower for patients having an aPTT ratio of 1.1-1.2 and 1.3-1.4 than those with an aPTT ratio <1.1.
This study is well written. Nevertheless, in the introduction or in the discussion, it is worth referring to current studies relating to stroke associated with atrial fibrillation in patients treated with VKAs and DOACs, e.g. Powers et al, Wańkowicz et al.
Response: Thank you. We have added the two references in the revised manuscript. We hope that the revisions have clarified the issues and the overall quality has been improved.
References (in the revised manuscript)
- Wańkowicz P, Nowacki P, Gołąb-Janowska M. Risk factors for ischemic stroke in patients with non-valvular atrial fibrillation and therapeutic international normalized ratio range. Arch Med Sci. 2019 Sep;15(5):1217-1222.
- Paweł Wańkowicz, Jacek Staszewski, Aleksander Dębiec, Marta Nowakowska-Kotas, Aleksandra Szylińska, Iwona Rotter. Ischemic Stroke Risk Factors in Patients with Atrial Fibrillation Treated with New Oral Anticoagulants. J Clin Med. 2021 Mar 16;10(6):1223. doi: 10.3390/jcm10061223.